# Artificial Sporulation Induction (ASI) by *kinA* Overexpression Affects the Proteomes and Properties of *Bacillus subtilis* Spores

**DOI:** 10.3390/ijms21124315

**Published:** 2020-06-17

**Authors:** Zhiwei Tu, Wishwas R. Abhyankar, Bhagyashree N. Swarge, Nicole van der Wel, Gertjan Kramer, Stanley Brul, Leo J. de Koning

**Affiliations:** 1Laboratory for Molecular Biology and Microbial Food Safety, University of Amsterdam, Science Park 904, 1098 XH Amsterdam, The Netherlands; Z.Tu@uva.nl (Z.T.); w.r.abhyankar@gmail.com (W.R.A.); B.N.Swarge@uva.nl (B.N.S.); 2Laboratory for Mass Spectrometry of Biomolecules, University of Amsterdam, Science Park 904, 1098 XH Amsterdam, The Netherlands; g.kramer@uva.nl (G.K.); l.j.dekoning@uva.nl (L.J.d.K.); 3Department of Medical Biology, Electron Microscopy Centre Amsterdam, Amsterdam University Medical Centers, University of Amsterdam, 1100 DD Amsterdam, The Netherlands; n.n.vanderwel@amc.uva.nl

**Keywords:** *B. subtilis*, *kinA* induction, spore proteomics, sporulation homogeneity

## Abstract

To facilitate more accurate spore proteomic analysis, the current study focuses on inducing homogeneous sporulation by overexpressing *kinA* and assesses the effect of synchronized sporulation initiation on spore resistance, structures, the germination behavior at single-spore level and the proteome. The results indicate that, in our set up, the sporulation by overexpressing *kinA* can generate a spore yield of 70% within 8 h. The procedure increases spore wet heat resistance and thickness of the spore coat and cortex layers, whilst delaying the time to spore phase-darkening and burst after addition of germinant. The proteome analysis reveals that the upregulated proteins in the *kinA* induced spores, compared to spores without *kinA* induction, as well as the ‘wildtype’ spores, are mostly involved in spore formation. The downregulated proteins mostly belong to the categories of coping with stress, carbon and nitrogen metabolism, as well as the regulation of sporulation. Thus, while *kinA* overexpression enhances synchronicity in sporulation initiation, it also has profound effects on the central equilibrium of spore formation and spore germination, through modulation of the spore molecular composition and stress resistance physiology.

## 1. Introduction

Spore forming bacteria, such as *Bacillus subtilis*, develop into an endospore when environmental conditions are unfavorable. A sporulating cell asymmetrically divides into two compartments of different sizes; a smaller forespore and a larger mother cell. The forespore is, then, engulfed by the mother cell and develops into a mature spore. The mother cell provides a nurturing environment for spore development. At the end of sporulation, the spore is released coincidently with the lysis of the mother cell. Sigma factors, including SigF, SigE, SigG and SigK, become active during sporulation and the direct transcription of genes that result in spore formation [1,2]. Spo0A, the master transcription regulator of sporulation and activates two types of promoters; one is used by the RNA polymerases containing SigA, the other containing SigH [3]. The transcription of *sigF* in the forespore and *sigE* in the mother cell is activated by a certain amount of phosphorylated Spo0A (Spo0A~P) [4,5]. Meanwhile, *sigG* and *sigK* are separately and sequentially activated further in the forespore and mother cell, respectively [1]. SigA, the housekeeping sigma factor, stays active in both forespore and mother cell during the process of sporulation [6,7].

The initiation of sporulation takes place heterogeneously. It is controlled by a phosphorelay system, which is in charge of phosphorylating Spo0A. In this phosphorelay, the sporulation kinases (KinA, B, C, D and E) phosphorylate Spo0F which in turn phosphorylates Spo0B ultimately leading to Spo0A phosphorylation [1,8]. Once Spo0A~P is accumulated and reaches a threshold, it triggers sporulation [9]. The heterochronic activation of phosphorelay genes is a crucial cause of sporulation heterogeneity, and the overexpression of *kinA, B* and *C* results in the loss of this heterogeneity [10]. In the absence of *kinA* expression, *kinB* has the ability to rescue sporulation [11]. Additionally, fluctuation in the transfer of the phosphoryl groups through the phosphorelay forces the phosphorylation rates of Spo0A to differ between the sporulating cells [12]. Spo0A can only be phosphorylated in-between the DNA replication cycles [13]. The decrease of growth rates prolongs the time between two DNA replications, which allows Spo0A~P to reach the threshold, thereby triggering the onset of sporulation [14]. Further contributing to sporulation heterogeneity is the fact that the sporulating cells at the early stage of sporulation cannibalize their non-sporulating siblings to support their vegetative growth and, ultimately, their spore formation [15].

The initiation heterogeneity of sporulation makes sure that sporulation, an energy- and time-consuming process, remains the last-resort to resist unfavorable conditions. However, whether such heterogeneity is purposely inbuilt to modulate, for instance, the heterogeneity in spore germination, is not known. Sporulation and germination heterogeneity causes problems in bacterial spore control strategies of food industries [16]. From an analytical point of view, sporulation heterogeneity complicates proteomic studies of spore protein composition, as the level of homogeneity of a sample determines how well protein analyses can be coupled to certain spore developmental stages. To facilitate a more sensitive spore proteomic analysis, a *kinA*-inducible strain of *Bacillus subtilis* may be used to synchronize input of phosphoryl groups and trigger sporulation [12,14,17,18]. However, how well sporulation can be synchronized and what the effect is, of the synchronization by overexpressing *kinA* on the spore properties, is not clear.

The current study focuses on synchronizing the initiation of sporulation by overexpressing *kinA*, thus synchronizing the expression of KinA, and controlling the medium glucose concentration, where high glucose represses, and low glucose derepresses, the phosphorylation of Spo0A. In addition, we assess the effect of the artificial sporulation induction (ASI) on spore resistance, the germination behavior at single-spore level and the spore structure. Finally, the proteomes of spores with and without *kinA* overexpression were assessed. The results indicate that synchronized sporulation by overexpressing *kinA* can generate a spore yield of 70% within 8 h. The procedure increases spore wet heat resistance and thickness of the spore coat and cortex layers, whilst delaying the time to spore germination (phase-darkening) and burst during spore outgrowth. The proteome analysis reveals that the upregulated proteins in the *kinA*-induced spores, compared to spores without *kinA* induction as well as the ‘wildtype’ spores, are mostly involved in spore formation. The downregulated proteins mostly belong to the categories associated with coping with stress, carbon and nitrogen metabolism, as well as the regulation of sporulation. Thus, while *kinA* overexpression through ASI enhances the synchronicity in sporulation initiation, it also has profound effects on the central equilibrium of spore formation and spore germination through the modulation of the spore molecular composition and stress resistance physiology.

## 2. Results

### 2.1. Overexpression of kinA Minimizes Heterogeneity in Sporulation Initiation

A concentration series of IPTG (Isopropyl β-D-1-thiogalactopyranoside) was tested and induction with 100 μM IPTG was selected for subsequent experiments (Appendix A). As shown in Figure 1A, enhanced *kinA* expression through ASI yields 70% spores within 8 h, while in that timeframe, only a small portion of spores is formed in the wildtype and uninduced mutant cells. Given the generally accepted notion on the time needed to form spores (see, for example, Ref. [9,10]) and this high 8 h yield, we inferred that ASI spore formation was markedly synchronized (see also microscopic images for t = 3, 4, 5, 6 and 7 h in Appendix A). A further 10% of the cells sporulated in the next 16 h to yield a maximum of 80% spores in the induced culture after 24 h. This yield is comparable to the non-synchronously initiated sporulation achieved in the wildtype strain within 24 h. The addition of IPTG does not affect the spore yield of the wildtype strain. Uninduced mutant cells yield around 50% spores after 24 h, presumably mediated by *kinB*, which is still present in the mutant cells. Their corresponding microscopic images are shown in Figure 1B.

### 2.2. Synchronized Sporulation Yields Spores with a Thicker Cortex, Coat Layers and a Larger Spore Diameter

Over 50 cross-sectioned images for every M+/− and WT+/− spores were taken using a transmission electron microscope (TEM). The thickness of coat and cortex layers and diameter of spores and their cores are measured. Figure 2A shows different spore structures. The mean thickness of cortex and coat layers of M+ spores are 72.5 ± 13.8 nm and 92 ± 17.9 nm, respectively. Both layers are significantly thicker than M− (cortex, 49.5 ± 9.5 nm; coat, 83.8 ± 13.7 nm) and WT+/− spores (WT+ cortex, 41.4 ± 8.0 nm, WT+ coat, 83.4 ± 11 nm; WT− cortex, 46.7 ± 10.3 nm, WT− coat, 81.8 ± 16.4 nm) (Figure 2B). The M+ spore core (355.9 ± 57 nm) is significantly larger than WT+ spores (331.9 ± 35.6 nm). The M+ spore diameter (773 ± 80 nm) is significantly larger than M− spores (685.1 ± 71.8 nm) and WT+/− spores (WT+, 648.5 ± 38.9 nm; WT−, 672.1 ± 49.5 nm) (Figure 2C).

### 2.3. Spores upon Synchronously Initiating Sporulation Acquire Increased Wet Heat Resistance and Are Delayed in Start of Germination and Burst

From the spores treated at 85 °C. Figure 3 shows that more M+ spores have survived the wet heat treatment of 10 and 15 min compared to WT+/− spores, but no significant difference is evident in the number of surviving spores following 20 min of wet heat treatment. The time required for the start of spore germination and the burst time have been analyzed for individual spores. M+ spores need a significant longer time (27.3 ± 12.3 min) to start germination compared to WT+/− and M− spores (9.3 ± 4.3 min, 11.2 ± 4.9 min and 9.8 ± 3 min respectively; Figure 4A). Moreover, while there is no difference between M+ and M- spores in terms of time from the end of germination to the burst, the M+ and M− spores need a significant longer time (183.9 ±104.9 min and 187.5 ± 70.7 min) to burst compared to WT+ and WT− spores (137 ± 36.5 min and 129.3 ± 25.8 min; Figure 4B).

### 2.4. Most Upregulated Proteins and a Subgroup of Downregulated Proteins in M+ Spores Are Related to Sporulation

The proteome comparisons of M+ spores to the M− and WT+ spores has led to the quantification of 631 and 840 proteins respectively (Appendix A). Of the quantified proteins, 109 (for M+ vs. M− spore comparison) and 135 (for M+ vs. WT+ spore comparison) proteins are differentially expressed (Figure 5A,B). SigA, SigE, SigK, SigG, GerE and SpoIIID are the transcriptional regulators of the genes of the upregulated proteins (Figure 6A,B). Sporulation is the category where most of the upregulated proteins are classified into, and a comparable number of downregulated proteins are also classified as being involved in sporulation (Figure 6C,D). The differentially expressed proteins related to sporulation are listed in Table 1. KinA itself is as expected induced in M+ spores, represented by the higher log_2_ transformed protein ratios of 4.11 (M+/M−) and 5.29 (M+/WT+) for KinA. These essentially indicate that the M+ spores have 17 to 39 times the amount of KinA protein compared to M− and WT+ spores. Among the upregulated sporulation proteins, CotQ, CotU, YheC, YjqC, CotC, CotJABC, SpsB, YabG and GerT are coat proteins; YdhD is one of the spore cortex lytic enzymes [19], and the others are proteins of unknown function. The downregulated sporulation proteins include coat proteins CgeA, CotG, CotW, CotX, small acid-soluble spore proteins (SASPs) SspA, SspE, SspG proteins OppA, OppC, OppD, OppF, GlnG, ParA, PbpF, YuaG, BdbD, DacB, PhoP, PhoR, SpoVD, and some proteins of unknown function. MurG, involved in cortex peptidoglycan precursor synthesis, is upregulated, albeit, N-acetylmuramic acid deacetylase PdaA, crucial in cortex formation, is downregulated. GerBC, one of the germinant receptors (GRs), is downregulated in M+ spores.

Supplementary to sporulation related proteins; most downregulated proteins fall into the category of carbon and amino acid/nitrogen metabolism as well as coping with stress (Figure 6C,D, Appendix A). Their primary transcriptional regulators are SigA, CcpA, CodY and AbrB (Figure 6A,B). The proteins of carbon and amino acid/nitrogen metabolism are mostly associated with the transfer, utilization and biosynthesis of amino acids, peptides and other small molecules.

No known sporulation related proteins are affected by the addition of IPTG to the wildtype sporulating cells. Two proteins are affected in the comparison of WT+/− spores (Appendix A). YhzC, a functionally unknown protein, is downregulated, and TasA, the major component of biofilm matrix [20], is upregulated.

### 2.5. Abolishing KinA Synthesis Has Profound Influence on the Proteome of Vegetative Cells and Spores

When IPTG is not added, the vegetative cells of the mutant strain have no KinA expressed. Amid 960 quantified proteins in the comparison of M− and WT− cells, 46 are differentially expressed, and in fact, 45 of them are downregulated in M- cells (Appendix A, Appendix A). SigA, AbrB, SigB, PhoP and CcpA are the primary transcriptional regulators of the downregulated proteins (Appendix A). It also appears that coping with stress is the major functional category of the downregulated proteins (Appendix A, Appendix A). Two proteins YraF (Log_2_(M−/WT−) = −4.29) and BdbD (Log_2_(M−/WT−) = −1.06) are related to sporulation. YraF is a forespore-specific sporulation protein and BdbD is a thiol-disulfide oxidoreductase.

M− spores (compared with WT− spores) have 28 differentially expressed proteins out of 725 quantified proteins (Appendix A, Appendix A). Primary transcriptional regulators are SigA and CodY (Appendix A). Five proteins, CgeA (Log_2_(M−/WT−) = 2.64), YbbC (Log_2_(M−/WT−) = −1.01), OppA (Log_2_(M−/WT−) = −1.45), OppD (Log_2_(M−/WT−) = −1.57) and OppF (Log_2_(M−/WT−) = −1.40), are related to sporulation. CgeA plays a role in maturation of the crust layer of the spore, while OppA, OppD and OppF are associated with the initiation of sporulation and competence development. YbbC is a protein of unknown function. Amino acid/nitrogen metabolism is the main functional category of the differentially expressed proteins (Appendix A, Appendix A).

## 3. Discussion

In the current set of experiments, *B. subtilis* gene *kinA* has been artificially induced to synchronize the expression of KinA in the phosphorelay regulating spore formation, while the glucose concentration in the environment is kept at 40mM (commonly used 10 mM), to halt the initiation of sporulation due to catabolite repression on Spo0A phosphorylation induced by high glucose concentration [21]. Sporulation initiation soon follows upon the dilution of glucose. This ASI resulting in the synchronization of sporulation initiation results in the formation of a high percentage of spores in a short time (70% spores within 8 h), but does not alter the final yield of 80% spores, which is comparable to the wildtype strain used as a reference. For uninduced mutant cells, 50% spores are formed at 24 h, where other kinases (mainly KinB) may trigger sporulation in the absence of KinA.

We observe that, due to the IPTG mediated induction of KinA, *B. subtilis* spore properties are affected. The increase in thickness of the spore coat and cortex leads to the spore having a larger diameter. The ultrastructural change in the spore coat and cortex morphology might affect spore resistance to unfavorable conditions. This is corroborated by the fact that a higher number of spores obtained via *kinA*-induction survive the wet heat treatment, as compared to the non-induced or WT spores. The proteomic comparison reveals that several spore coat proteins (CotQ, CotU, CotC, CotJABC) are upregulated. However, CotG, CotW and CotX are downregulated in spores from *kinA* induced cell cultures. Coat protein CgeA is downregulated in M+ spores compared to WT+ spores and is upregulated in M− spores compared to WT− spores. Clearly, KinA is highly overexpressed in M+ spores versus M− spores, which illustrates that this downregulation of CgeA in M+ spores is independent of the overexpression of KinA. The same observation is made in the case of OppA, OppD, OppF and YccB proteins. The upregulation of CgeA in M− spores suggests that not all of the coat proteins are downregulated due to the absence of KinA. The upregulated coat proteins (for example CgeA) in M− spores may relate to the higher number of colonies formed from M− spores compared to WT+/− spores after wet heat treatment. Almost all the upregulated proteins in M+ spores are related to sporulation, focusing on the spore coat proteins and proteins relevant to the regulation of the biogenesis of the spore coat. In contrast, the downregulated sporulation proteins in M+ spores are mostly GerBC and sporulation proteins not related to spore coat formation. Other GRs were not quantified here. This could be due to the overall low abundance of GRs. GerBC was downregulated in spores originating from KinA overexpressing cells, in the comparison with WT+ spores. While this observation may not fully explain the delay of germination of M+ spores, it is known that downregulation in GRs is closely related to spores being less prone to germination [22,23]. The increased thickness of the spore cortex may correlate to a change in two quantified proteins, MurG and PdaA, where their quantitative behavior is opposite, MurG upregulated and PdaA downregulated. Both proteins are related to cortex formation. The absence of KinA also has a profound influence on the proteome of vegetative cells, indicating that KinA may have regulatory functions in vegetative cells.

Finally, metabolism in the forespore diminishes after completion of forespore engulfment before it ceases completely in the mature spores [24]. Data by Swarge et al. [25] show that spores are charged with a set of proteins during sporulation, needed for the basic metabolic action in the early stages of spore outgrowth to vegetative cells. We speculate that the fact that M+ and M− spores took more time to burst, i.e., to complete the initial phase of outgrowth, may have a relation with the downregulated proteins involved in cellular metabolism.

SigA has been demonstrated to play roles in germination, outgrowth and housekeeping functions during sporulation [7]. Our results suggest that SigA may be the top regulator of the differentially expressed proteins, apart from those proteins exclusive to sporulation, which are under the control of the sigma factors (SigE, SigF, SigK and SigG), in both vegetative cells and spores, upon varying the expression of KinA.

In conclusion, while *kinA* overexpression enhances synchronicity in sporulation initiation, it also leads to an increase in thickness of spore cortex and coat, an increase in thermal resistance, and a delay in germination and burst (outgrowth). These data support the notion that KinA has effects on the spore formation, as well as the spore molecular and physiological prosperity. The trade-off is between survival and stress resistance, versus the rate of germination and the growth of the population. For all spore forming organisms, the balance between both developmental states is central to their evolutionary success, where an extreme stress resistance at the expense of extremely low outgrowth rate/efficiency ultimately may be seen as equivalent to cell death. Further studies on the links of the KinA mediated phosphorelay with cellular physiology are needed for an in-depth understanding of the regulation of spore formation and spore resistance.

## 4. Materials and Methods

### 4.1. Bacterial Strain and Sporulation

*Bacillus subtilis* PY79 (wildtype, WT) and its derivative strain 1887 (*kinA*-inducible mutant, M, obtained from Prof. M. Fujita’s lab, University of Houston) were used in this study [26]. The native *kinA* of strain 1887 is fused to an IPTG inducible promoter *P*_hyper-spank_. In the absence of IPTG, no *kinA* is expressed in the mutant cells. Cells of either the wildtype or mutant strain were cultured in a MOPS buffered medium at 37 °C, 200 rpm, to generate spores [27]. Cells from a single colony on an agar LB plate were grown in a MOPS buffered medium, with 40 mM glucose and 40 mM NH_4_Cl, until an OD_600_ value of ~0.65 was reached. Final various concentrations between 0–200 µM IPTG were added to the culture and continued for further 90 min. It was then diluted six times with the same MOPS buffered medium, pre-warmed, but without glucose to dilute the glucose 6 times. This final culture was incubated for 24 h to induce sporulation.

### 4.2. Sporulation Efficiency and Sample Harvesting

Sporulating mutant cells induced with various concentrations of IPTG were harvested at 8 h after glucose dilution, to calculate sporulation efficiency according to a previous protocol [28]. Then, a fixed concentration of IPTG was selected for the rest of the experiments. Sporulating cells for wildtype and mutant strain, with or without IPTG, were harvested at 8 and 24 h after glucose dilution, to calculate sporulation efficiency. Three biological replicates were performed for each experiment. Vegetative cell samples were harvested before glucose dilution and stored at −80 °C. Spore samples were harvested 24 h after glucose dilution and purified from remaining vegetative cells with Histodenz (Sigma-Aldrich, St. Louis, Missouri, USA) density gradient centrifugation, as described previously [29]. Purified spores were stored at −80 °C.

### 4.3. Electron Microscopy

One batch of the wildtype spores, with and without IPTG addition (WT+ and WT− spores), and mutant spores with and without *kinA*-induction (M+ and M− spores) were fixed in 1% glutaraldehyde and 4% paraformaldehyde in 0.1 M Phosphate buffer (pH 7.4) for 24 h. After fixation, the spores were washed in distilled water, block stained overnight in 1.5% uranyl acetate to enhance contrast in the electron microscope, washed in distilled water, and osmicated for 60 min in 1% OsO_4_ (Electron Microscopy Sciences, Hatfield, PA, USA) in water. Subsequently, the spores were dehydrated in an alcohol series (of ethanol) and embedded into propylene oxide/Epon 1:1, before embedding into LX-112 resin (Ladd Research, Williston, VT, USA). Between each step, the spores were centrifuged down. After polymerization at 60 °C, the Epon blocks with the spores were prepared for ultrathin sectioning. Ultrathin sections of 90 nm were cut on a Reichert EM UC6 with a diamond knife, collected on Formvar coated grids and stained with uranyl acetate and lead citrate. Sections were examined with a FEI Technai12 G2 Spirit Biotwin transmission electron microscope (TEM). Images were taken with a Veleta camera (Olympus, SIS, Münster, Germany). For measurement of the thickness of the coat layers, using the iTEM software (Emsis, Münster, Germany), the images of cross-sectioned spores were taken at a magnification of 68,000×. For these measurements, more than 50 spores for every WT+, WT−, M+ and M− sample were used. The thickness of different spore layers is an average of four measurements at different positions on the rectangular cross-sections for each spore and for each layer. The diameters of spores and spore cores are the averages of two measurements across horizontal and vertical axes.

### 4.4. Live Imaging of Spores

Germination and bursts of spores were observed with the Nikon Eclipse Ti. The Nikon Eclipse Ti had for phase contrast imaging a Prior Brightfield LED, a Nikon CFI Plan Apo Lambda 100X Oil, C11440-22CU Hamamatsu ORCA flash 4.0 camera, LAMBDA 10-B Smart Shutter from Sutter Instrument, an OkoLab stage incubator, and was equipped with NIS elements software version 4.50.00. The slide was prepared according to the method developed in our laboratory [30]. In brief, a 1.5 cm × 1.6 cm Gene Frame^®^ (Thermo Scientific, Landsmeer, Netherlands) was attached to a microscope slide with a semisolid matrix pad, made of 1.5% agarose with complete minimal medium [31], 10 mM L-valine and AGFK (10 mM L-asparagine, 10 mM glucose, 1 mM fructose and 1 mM potassium chloride), in the centre of the frame. Spores were then loaded on the pad and covered with a 18 × 18 mm glass coverslip (Thermo Scientific). Pressure was applied on the edges of the cover slip onto the Gene Frame^®^, to make it a closed air-containing chamber which was then used for time-lapse microscopy. A sample frequency of 1 frame per 2 min was taken for the individual spores. The start of germination, end of germination and burst of each spore from three biological replicates were assessed with the aid of the ImageJ plugin SporeTrackerX [32]. 

### 4.5. Heat Resistance of Spores

Heat resistance test of spores was carried out following a previously published protocol [33]. In addition, spores were treated at 85 °C for 0, 10, 15 and 20 min after heat activation of 70 °C and non-heat-treated spores used as a control. Viable spores were estimated by counting CFU’s on LB agar plates after 48 h. Three biological replicates were performed.

### 4.6. Sample Preparation for LCMS Analysis

Here, a ^15^N metabolic labelling strategy was used to perform proteome comparison [34]. One sample prepared as described above with ^14^NH_4_Cl as a sole nitrogen source was mixed with another sample, which was prepared accordingly, but ^14^NH_4_Cl was replaced by ^15^NH_4_Cl to perform the proteomic comparison. In this study: (1) the spores of wildtype and mutant strain, both with IPTG induction (WT+ and M+ spores), were compared to address the proteome difference between normal spores and spores from synchronized sporulation, in which *kinA* is overexpressed; (2) the spores of mutant strain with and without IPTG induction (M+ and M- spore) were compared to address the effect of no expression and overexpression of *kinA*; (3) the spores of wildtype strain with and without IPTG induction (WT+ and WT− spores) were compared to address the effect of addition of IPTG to wildtype spores. (4) The vegetative cells of the wildtype and mutant strains, both without IPTG induction (WT− and M− cells), were compared to address the cell proteome differences, owing to the expression and no-expression of *kinA*; (5) the spores of wildtype and mutant strain both without IPTG induction (WT− and M− spores) were compared to address the spore-proteome difference between expression and no-expression of *kinA*. Three biological replicates for every comparison were analyzed. Cells or spores from ^14^N and ^15^N medium were mixed in a 1:1 ratio based on OD_600_. The mixed sample was subjected to one pot sample processing and the fractionation of peptides using a HPLC system packaged with a SeQuant^®^ ZIC^®^-HILIC (5 μm, 200 Å) PEEK 100 mm × 2.1 mm HPLC column [35]. The gradient applied was from 0.05% formic acid in 90% acetonitrile, 9.95% Water to 0.05% formic acid-in 25% acetonitrile, 74.95% Water in 45 min at a flow rate of 400 µl/min, whilst collecting 10 fractions throughout the gradient.

### 4.7. LCMS Analysis

ZIC-HILIC fractions were re-dissolved in 0.1% TFA and peptide concentrations were determined by measuring the absorbance at wavelength of 205 nm [36]. LC-MS/MS data were acquired with an Apex Ultra Fourier transform ion cyclotron resonance mass spectrometer (Bruker Daltonics, Bremen, Germany), equipped with a 7 T magnet and a Nano electrospray Apollo II Dual Source coupled to an Ultimate 3000 (Dionex, Sunnyvale, CA, USA) HPLC system. Samples containing ~ 300 ng of the tryptic peptide mixtures were injected in 10 μL of 0.1% TFA, 3% ACN aqueous solution, together with 75 fmol of [Glu1]-Fibrinopeptide B human peptide and loaded onto a PepMap100 C18 (5 μm particle size, 100 Å pore size, 300 μm inner diameter × 5 mm length) precolumn. Following injection, the peptides were eluted onto an Acclaim PepMap 100 C18 (3 μm particle size, 100 Å pore size, 75 μm inner diameter × 250 mm length) analytical column (Thermo Scientific, Etten-Leur, the Netherlands) to the Nano electrospray source. Gradients ran from 0.1% formic acid-3% ACN, 96.9% water to 0.1% formic acid-50% ACN, 49.9% water in 120 min at a flow rate 300 nl/min. Data-dependent Q-selected peptide ions were fragmented in the hexapole collision cell at an argon pressure of 6 × 10^-6^ mbar (measured at the ion gauge) and the fragment ions were detected in the ICR cell at a resolution of up to 60,000. In the MS/MS duty cycle, three different precursor peptide ions were selected from each survey MS. The MS/MS duty cycle time for one survey MS and three MS/MS acquisitions was approximately 2s. Instrument mass calibration was better than 1 ppm over the range of 250–1500 m/z.

### 4.8. Data Processing and Statistics

The 10 ZIC-HILIC fractions were jointly processed as a multifile with MASCOT DISTILLER (version 2.4.3.1, 64 bits), MDRO 2.4.3.0 (MATRIX science, London, UK), including the Search toolbox and the Quantification toolbox. Peak-picking for both MS and MS/MS spectra were optimized for the mass resolution of up to 60,000. Peaks were fitted to a simulated isotope distribution, with a correlation threshold of 0.6, with minimum signal to noise ratio of 1.3. The processed data were searched in a MudPIT approach with the MASCOT server program 2.3.02 (MATRIX science, London, UK), against the *B. subtilis* 168 ORF translation database. The MASCOT search parameters were as follows: enzyme—trypsin, allowance of two missed cleavages, fixed modification—carbamidomethylation of cysteine, variable modifications—oxidation of methionine and deamidation of asparagine and glutamine, quantification method—metabolic ^15^N labelling, peptide mass tolerance and peptide fragment mass tolerance—50 ppm. A MASCOT MudPIT peptide identification threshold score of 20 and FDR of 2% were set to export the reports. Using the quantification toolbox, the quantification of the light peptides relative to the corresponding heavy peptides was determined as ^14^N/^15^N ratio, using Simpson’s integration of the peptide MS chromatographic profiles for all detected charge states. The quantification parameters were: correlation threshold for isotopic distribution fit—0.95, ^15^N label content—99.6%, XIC threshold—0.1, all charge states on, max XIC width—120 s, elution time shift for heavy and light peptides—20 s. All isotope ratios were manually validated by inspecting the MS spectral data. The protein isotopic ratios were then calculated as the average over the corresponding peptide ratios.

Log_2_ transformed protein isotopic ratios from three replicates were fitted to a linear model using R/Bioconductor software package *limma*, and statistical significance was assessed by performing empirical Bayes statistics [37]. The false discovery rate was controlled using the Benjamini–Hochberg procedure for false discovery rate [38]. All protein isotopic ratios were averaged ratios from at least two replicates out of three, unless otherwise stated. Proteins with a *p*-value less than 0.01 and an averaged log_2_ transformed protein ratios smaller than −1 or larger than 1 were considered to be differentially expressed. Differentially expressed proteins were categorized, and their transcriptional regulators were determined, according to SubtiWiki (http://subtiwiki.uni-goettingen.de/) [39].

## Figures and Tables

**Figure 1 ijms-21-04315-f001:**
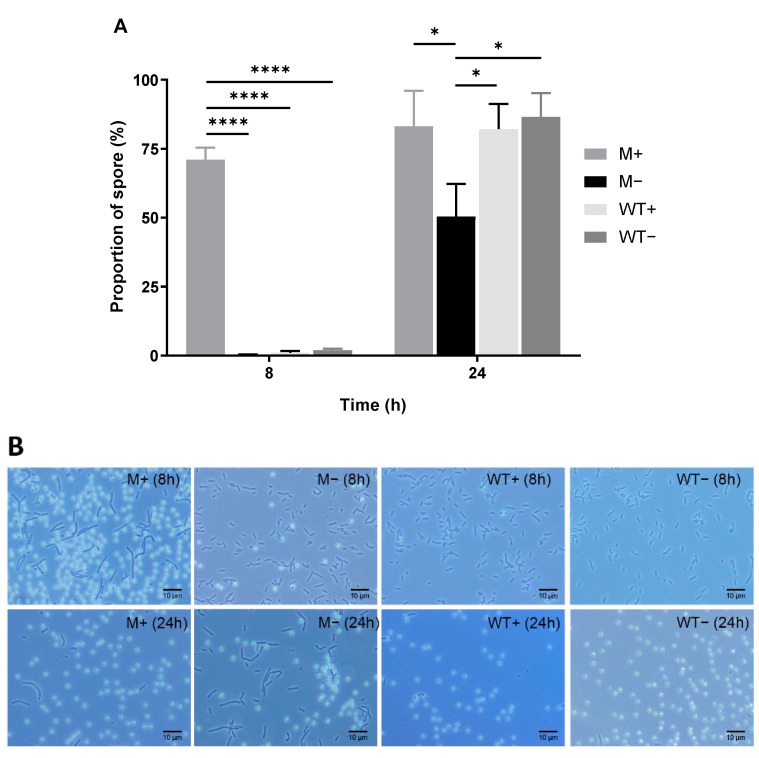
Proportion of spores of wildtype and mutant strain with and without IPTG, at 8 and 24 h after glucose dilution. M+, IPTG-induced spores. M−, uninduced spores. WT+, wildtype spores with addition of IPTG. WT−, wildtype spores without addition of IPTG. (**A**) Bar graph of the sporulation efficiency. Statistical significance is determined by one-way analysis of variance (ANOVA). *, *p* < 0.05; ****, *p* < 0.0001. (**B**) Typical microscopic images corresponding to (**A**).

**Figure 2 ijms-21-04315-f002:**
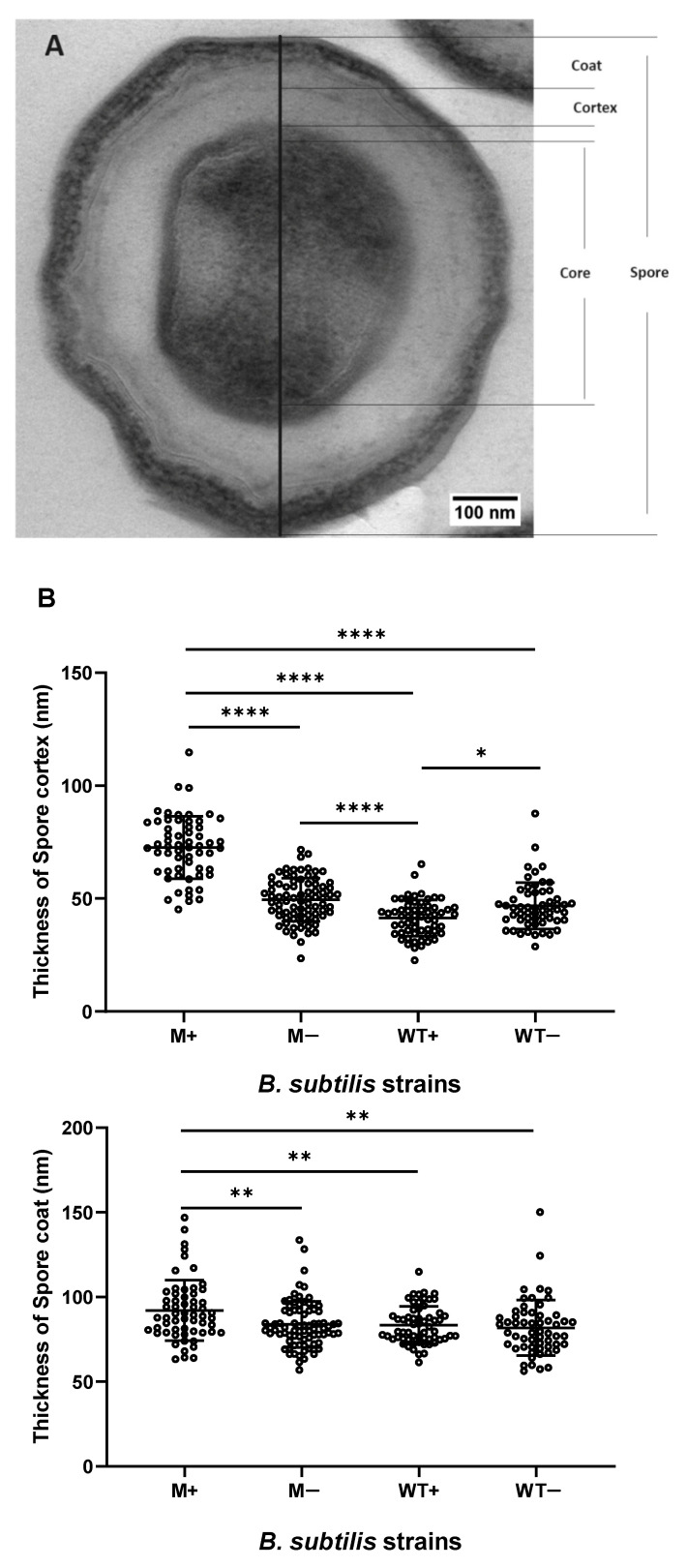
Effect of *kinA*-induction on spore structure of *B. subtilis* wildtype and *kinA*-inducible mutant strains. Provided are: a representative electron microscopic image of induced mutant spores (**A**) and the measurements of the thickness of spore coat and cortex layers (**B**) and the diameters of spores and their cores (**C**) from *Bacillus subtilis* wildtype and mutant strain, with and without IPTG. For every sample, over 50 individual spores have been measured (n_M+_ = 59, n_M−_ = 80, n_WT+_ =60, n_WT−_ = 57). Statistical significance is determined by a one-way analysis of variance (ANOVA). *, *p* < 0.05; **, *p* < 0.01; ****, *p* < 0.0001. Typical TEM images of different strains are submitted in Appendix A.

**Figure 3 ijms-21-04315-f003:**
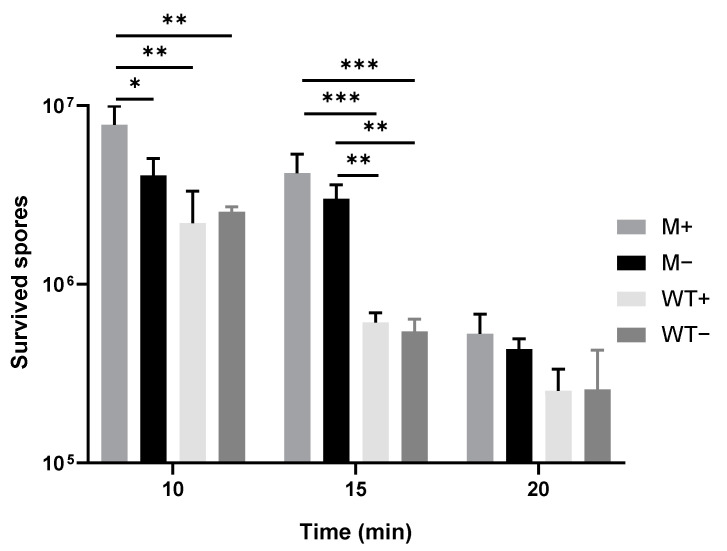
Effect of *kinA*-induction on spore survival after heat treatment. Statistical significance is determined by one-way analysis of variance (ANOVA). *, *p* < 0.05; **, *p* < 0.01; ***, *p* < 0.001.

**Figure 4 ijms-21-04315-f004:**
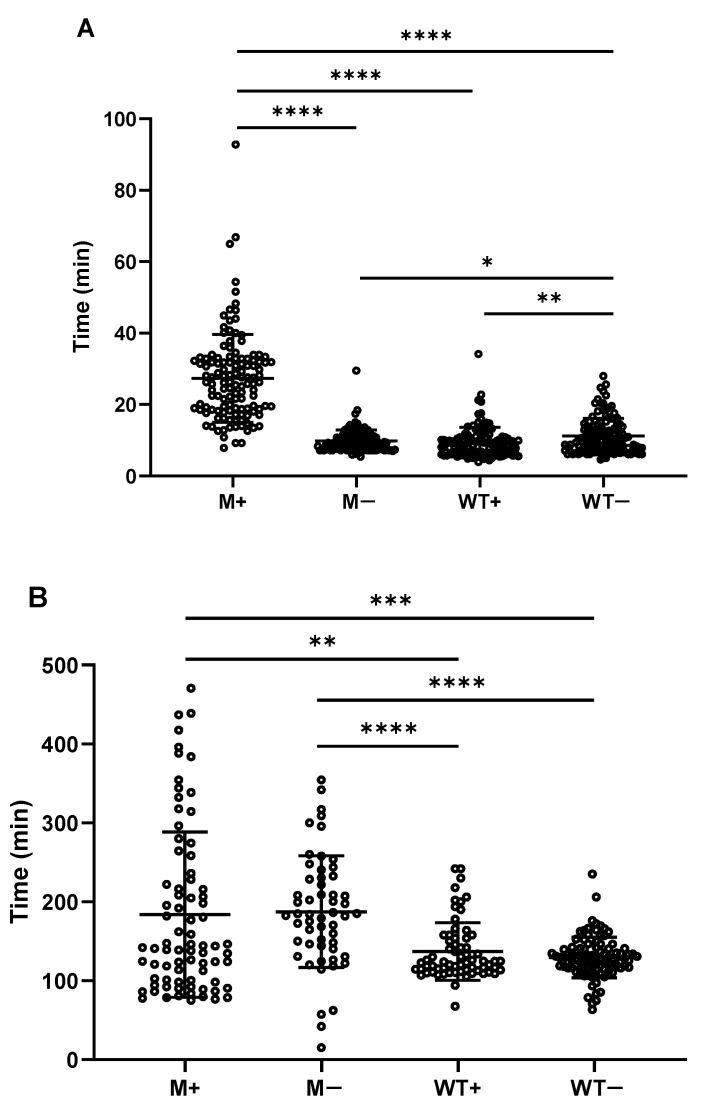
Distribution of time to start of germination and burst time of spores. Mean and standard deviation were shown in the distribution. (**A**) time to start of germination of M+ spores (*n* = 125), M−spores (*n* = 121), WT+ spores (*n* = 115) and WT− spores (*n* = 125), harvested at 24 h after glucose dilution. (**B**) time from end of germination to start of burst of spores for M+ (*n* = 78), M− (*n* = 51), WT+ (*n* = 66) and WT− (*n* = 96). Statistical significance is determined by one-way analysis of variance (ANOVA). *, *p* < 0.05; **, *p* < 0.01; ***, *p* < 0.001; ****, *p* < 0.0001.

**Figure 5 ijms-21-04315-f005:**
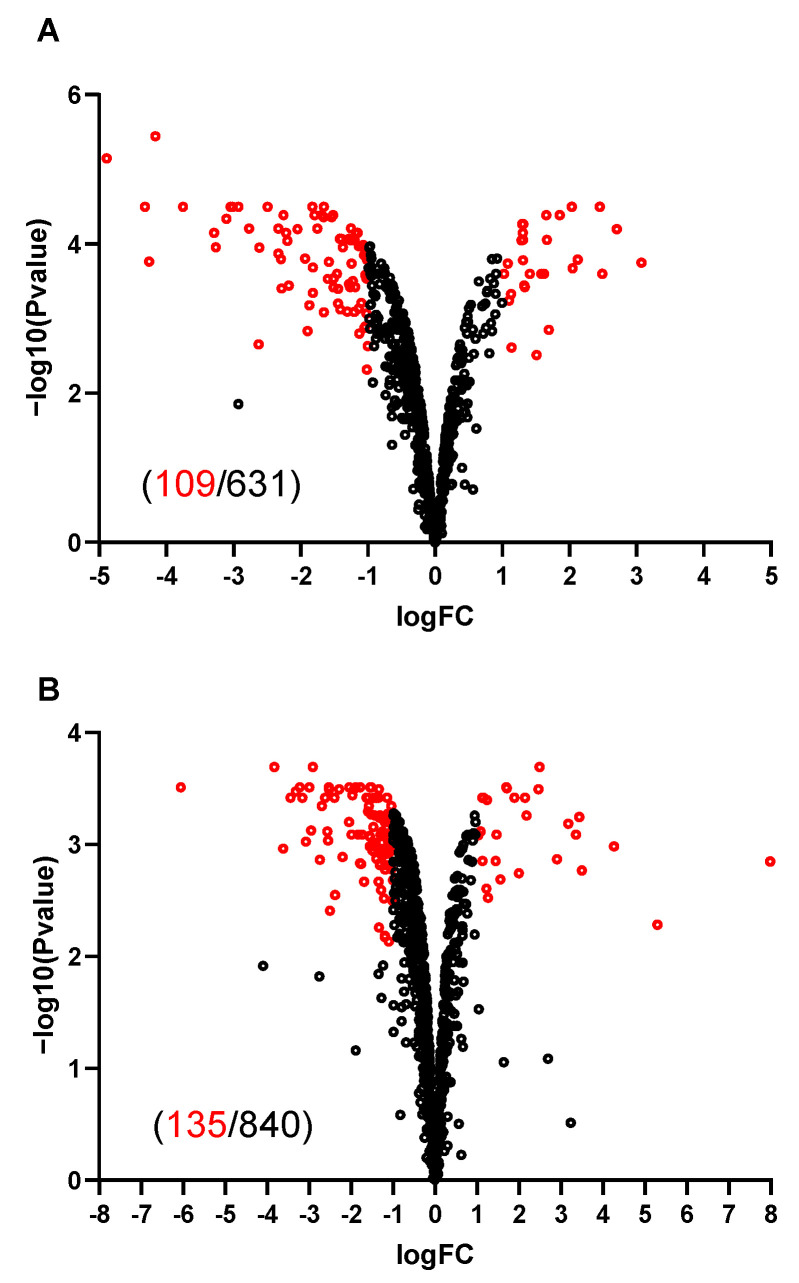
Volcano plots of proteomic comparison of spores. X-axis indicates the average of log2 isotopic ratios from the replicates. Negative values indicate downregulation and positive values indicate upregulation; Y axis is −log_10_
*p*-value. Dots in red indicate differentially expressed proteins. Dots in black indicate proteins that are not significantly changed in protein expression. Numbers in red and black in brackets are the number of differentially expressed proteins and total quantified proteins, respectively. (**A**) M+ spores are compared to M− sores. (**B**) M+ spores are compared to WT+ spores.

**Figure 6 ijms-21-04315-f006:**
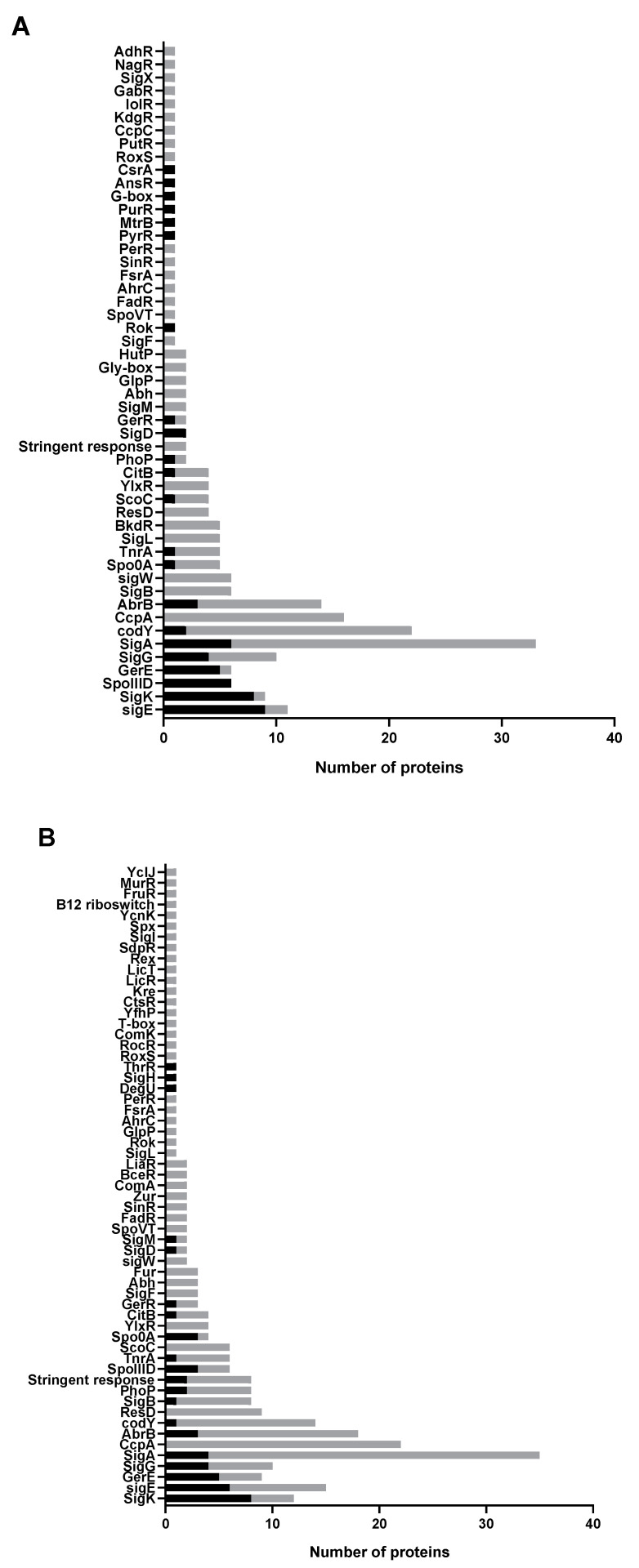
Regulators and functional classification of differentially expressed proteins. The X-axis indicates the number of proteins classified into regulatory or functional categories, as depicted on the Y-axis. Note that every protein could occupy more than one regulatory or functional category. Bars in black and gray indicate the number of upregulated and downregulated proteins, respectively. (**A**) Regulators of differentially expressed proteins in M+ spores compared to M− spores. (**B**) Regulators of differentially expressed proteins in M+ spores compared to WT+ spores. (**C**,**D**) are the corresponding functional classification to (**A**,**B**) and their respective comparisons.

**Table 1 ijms-21-04315-t001:** Differentially expressed sporulation related proteins in *kinA*-overexpressed mutant spores of *B. subtilis.*

Protein	Log_2_(M+/M−)	Log_2_(M+/WT+)	Function	Product
Upregulation				
CotQ	1.66	1.70	protection of the spore	spore coat protein
CotU	2.49	2.00	resistance of the spore	outer spore coat protein
GerE	2.70	2.90	regulation of SigK-dependent gene expression	transcriptional regulator (LuxR-FixJ family)
YheC	3.07	3.17	Unknown	ATP-binding spore coat protein
YhxC	2.03	1.24	Unknown	unknown
YjqC	1.66	2.49	protection of the spore	spore coat protein
YpqA	1.31	2.14	Unknown	unknown
YraF	1.41	1.09	Unknown	unknown
YtcC	2.05	2.46	lipopolysaccharide biosynthesis	sporulation protein
YxeD	1.29	1.47	Unknown	unknown
CotC	1.51	N.A.	resistance of the spore	spore coat protein (outer)
CotJA	1.31	N.A.	polypeptide composition of the spore coat	unknown
CotJB	1.30	N.A.	polypeptide composition of the spore coat	unknown
CotJC	1.32	N.A.	polypeptide composition of the spore coat, may protect against oxidative stress	putative manganese catalase
GerT	1.86	N.A.	germination	spore coat protein
YdhD	1.31	N.A.	Unknown	spore coat glycosylase
YqfT	1.03	N.A.	Unknown	unknown
YraG	1.33	N.A.	Unknown	unknown
KinA	4.11 ^a^	5.29	initiation of sporulation	two-component sensor kinase
MurG	N.A.	1.13	peptidoglycan precursor biosynthesis	UDP-N-acetylglucosamine-N-acetylmuramyl-(pentapeptide)pyrophosphoryl-undecaprenol N-acetylglucosamine transferase
SpsB	N.A.	1.71	spore coat polysaccharide synthesis	unknown
YabG	N.A.	1.03	modification of spore coat proteins	protease
YojB	N.A.	1.26	Unknown	unknown
YraD	N.A.	1.15	Unknown	unknown
Downregulation				
OppA	−4.16	−2.53	initiation of sporulation, competence development	oligopeptide ABC transporter (binding protein)
OppD	−2.93	−1.42	initiation of sporulation, competence development	oligopeptide ABC transporter (ATP-binding protein)
OppF	−3.04	−1.84	initiation of sporulation, competence development	oligopeptide ABC transporter (ATP-binding protein)
PdaA	−1.01	−1.12	spore cortex peptidoglycan synthesis	N-acetylmuramic acid deacetylase
SspA	−1.18	−1.59	protection of spore DNA	small acid-soluble spore protein (major alpha-type SASP)
YckD	−1.14	−1.43	Unknown	unknown
YugP	−1.02	−1.20	Unknown	unknown
CgeA	−3.27	N.A.	maturation of the outermost layer of the spore	spore crust protein
GlnH	−2.49	N.A.	glutamine uptake	glutamine ABC transporter (binding protein)
ParA	−1.01	N.A.	forespore chromosome partitioning/negative regulation of Sporulation initiation	negative regulator of Sporulation initiation
PbpF	−1.10	N.A.	bifunctional glucosyltransferase/transpeptidase	penicillin-binding protein 2C
YbfJ	−1.14	N.A.	Unknown	unknown
YhfN	−1.02	N.A.	Unknown	unknown
YuaG	−2.22	N.A.	involved in the control of membrane fluidity	membrane-associated scaffold protein
BdbD	N.A.	−1.40	oxidative folding of proteins	thiol-disulfide oxidoreductase
CotG	N.A.	−1.10	resistance of the spore	spore coat protein
CotW	N.A.	−1.34	resistance of the spore	spore crust protein (insoluble fraction)
CotX	N.A.	−1.31	spore crust assembly	spore crust protein (insoluble fraction)
DacB	N.A.	−1.06	carboxypeptidase	penicillin-binding protein 5*, D-alanyl-D-alanine carboxypeptidase
GerBC	N.A.	−1.28	germination	nutrient receptor
OppC	N.A.	−1.52	initiation of sporulation, competence development	oligopeptide ABC transporter (permease)
PhoP	N.A.	−1.55	regulation of phosphate metabolism (phoA, phoB, phoD, resABCDE, tagA-tagB, tagDEF, [tuaA-H])	two-component response regulator (OmpR family)
PhoR	N.A.	−1.93	regulation of phosphate metabolism	two-component sensor kinase
spoVD	N.A.	−1.30	spore morphogenesis	penicillin-binding protein (spore cortex)
SspE	N.A.	−1.09	Unknown	small acid-soluble spore protein (major gamma-type SASP)
SspG	N.A.	−1.01	protection of spore DNA	small acid-soluble spore protein (minor)
YbbC	N.A.	−1.32	Unknown	unknown
YdcC	N.A.	−1.30	Unknown	unknown
YjaZ	N.A.	−1.32	Unknown	unknown

^a^, is quantified in one replica. N.A.= not applicable; proteins were identified but not quantified.

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
