# Peer review of "Artificial Sporulation Induction (ASI) by kinA Overexpression Affects the Proteomes and Properties of Bacillus subtilis Spores"

_ijms, 2020, doi:10.3390/ijms21124315_

Round 1

Reviewer 1 Report

This is a nice proteomic study of Bacillus subtilis spore composition. The rationale behind the work is sound i.e. asynchronous sporulation of B. subtilis which leads to heterogeneous spore populations. Inducible expression of KinA is a logical attempt to try and improve this situation. This is achieved in terms of time to sporulate at the population level (70% within 8 hrs) although the final level is comparable to wild type spores (80%, meaning that 10% of the KinA spore population is presumably different in some ways to the bulk 70%). The proteomic work and related experiments are conducted to a high standard. However, I'm less convinced that the objective has been met since by most measures the KinA spores show greater heterogeneity than uninduced or WT spores! Plots in Figs 1, 2 and 4 exemplify this. Maybe this heterogeneity is accounted for by the 10% of spores that only sporulate after 8 hrs?

I have no major suggestions in terms of the manuscript, only the following typographical errors:

Line 37 - space between SigE and SigG

Line 49 - Heterochronic? I'm not sure what this means.

Line 77 - phase darkening and burst i.e. during spore germination (for the uninitiated).

Line 90 - delete While,

Figure 1 - I had no idea what M meant at this stage. Maybe add definition to legend.

Figure 2 - space between B. and subtilis (and elsewhere in figures)

Line 124 - from not form

Figure 3 Y-axis label - viable spores not Survived spores

Line 150 - YdhD is a CLE involve din spore germination (see Anne Moir's papers)

Line 151 - 154 - this sentence doesn't make sense.

Figure 5 legend - significantly

Line 190 - 192 - why should this be the case? Leaky expression? Polar effects?

Line 233 - seems out of context. GRs are mentioned din the preceding paragraph. Maybe 'only GerBC of GR proteins' for context.

Line 239 - it not if

Oine 252 - delete the before evolutionary

Line 264 - from not form

Line 271 - of IPTG

Line 284 - Why the company address here? What's the product?

Line 310 treated at not with

Author Response

Concerns: rebuttal Induced homogeneous sporulation by kinA-overexpression leads to altered Bacillus subtilis spore proteomes and spore properties. IJMS-802450 (see also attachment).

 Dear editor,

I would hereby like to submit our rebuttal and amended manuscript. We would like to answer the points put forward by the reviewers as follows.

First, we agree with both reviewers that the title did not optimally match the content of the manuscript. Hence, we have revised the title to: Artificial sporulation induction (ASI) by KinA overexpression affects the proteomes and properties of Bacillus subtilis spores.

Below we answer the specific points put forward by reviewer 1, whom we thank for the constructive criticism.

Reviewer 1

This is a nice proteomic study of Bacillus subtilis spore composition. The rationale behind the work is sound i.e. asynchronous sporulation of B. subtilis which leads to heterogeneous spore populations. Inducible expression of KinA is a logical attempt to try and improve this situation. This is achieved in terms of time to sporulate at the population level (70% within 8 hrs) although the final level is comparable to wild type spores (80%, meaning that 10% of the KinA spore population is presumably different in some ways to the bulk 70%). The proteomic work and related experiments are conducted to a high standard. However, I'm less convinced that the objective has been met since by most measures the KinA spores show greater heterogeneity than uninduced or WT spores! Plots in Figs 1, 2 and 4 exemplify this. Maybe this heterogeneity is accounted for by the 10% of spores that only sporulate after 8 hrs?

Our reply: In agreement with the reviewer, we cannot rule out interference of the remaining 10% spore population regarding the final, at 24 h, observed spore homogeneity. In addition, in Fig. 4 please note that germination behavior of different strains is shown. KinA overexpression causes downregulation of GerBC (Table 1 of the manuscript), one of the major germination receptor protein subunits. This may result in the observed delay in germination for different spores from same population and thus result at the population level in heterogeneity in spore germination. KinA overexpression also increases spore thermal resistance. Spores with lower germinant receptor levels as well as more heat resistant spores tend to show a higher heterogeneity in germination and outgrowth (see for instances Krawczyk A.O. et al. 2017. Spore Heat Activation Requirements and Germination Responses Correlate with Sequences of Germinant Receptors and with the Presence of a Specific spoVA2mob Operon in Foodborne Strains of Bacillus subtilis. Appl. Env. Microbiol. 83, 7, e03122-16).

I have no major suggestions in terms of the manuscript, only the following typographical errors:

Line 37 - space between SigE and SigG

Our reply: Done

Line 49 - Heterochronic? I'm not sure what this means.

Our reply: Heterochronic indicates that the phosphorelay gene expression is highly dynamic or “heterochronic” between individual cells.

Line 77 - phase darkening and burst i.e. during spore germination (for the uninitiated).

Our reply: Added.

Line 90 - delete While,

Our reply: Done.

Figure 1 - I had no idea what M meant at this stage. Maybe add definition to legend.

Our reply: We expanded the legend of figure 1. Line 103-107.

Figure 2 - space between B. and subtilis (and elsewhere in figures)

Our reply: Done.

Line 124 - from not form

Our reply: Done.

Figure 3 Y-axis label - viable spores not Survived spores

Our reply: Done.

Line 150 - YdhD is a CLE involved in spore germination (see Anne Moir's papers)

Our reply: Text was amended to include the notion that YdhD is one of spore cortex lytic enzymes.

Line 151 - 154 - this sentence doesn't make sense.

Our reply: The text was changed to “the downregulated sporulation proteins include coat proteins CgeA, CotG, CotW, CotX, small acid-soluble spore proteins (SASPs) SspA, SspE, SspG and others.”

Figure 5 legend – significantly

Our reply: Done.

Line 190 - 192 - why should this be the case? Leaky expression? Polar effects?

Our reply: We cannot rule out polar effects. Regarding leaky expression, based on the article, ‘Eswaramoorthy P, Duan D, Dinh J, Dravis A, Devi SN, Fujita M. The threshold level of the sensor histidine kinase KinA governs entry into sporulation in Bacillus subtilis. J Bacteriol. 2010;192(15):3870‐3882. doi:10.1128/JB.00466-10’, where the mutant was extensively used, see also ‘Fujita M. and Losick, R. Evidence that entry into sporulation in Bacillus subtilis is governed by a gradual increase in the level and activity of the master regulator Spo0A. Genes & Development 2005; 19: 2236-2244  doi/10.1101/gad.1335705’, where the mutant was originally described, said mutation is leak proof. Therefore, we also assume the same. Finally, KinA overexpression affects many regulons as shown in the results. Therefore, there may be yet unforeseen regulatory effects. More detailed studies are needed to unravel.  

Line 233 - seems out of context. GRs are mentioned in the preceding paragraph. Maybe 'only GerBC of GR proteins' for context.

Our reply: The text has been amended to improve the flow of information discussion.

Line 239 - it not if

Our reply: Done.

Line 252 - delete the before evolutionary

Our reply: Done.

Line 264 - from not form

Our reply: Done.

Line 271 - of IPTG

Our reply: Done.

Line 284 - Why the company address here? What's the product?

Our reply: The text was amended. The company name refers to the source of OsO4.

Line 310 treated at not with

Our reply: Done.

Reviewer 2 Report

GENERAL REMARKS

the purpose of this study is not very clear. Is it to achieve synchronized sporulation or to analyze the consequences of over-expression or inhibition of kinA expression? Consequently, the entire organization of this article should be reviewed to ensure that the objective is clearly defined and analyzed throughout the document.

In addition, in my opinion, essential information is missing (see below various remarks in the "Result" section). As a result, the interpretation of some results can be controversial.

in conclusion, I consider that this article is not acceptable as it stands to be published in IJMS.

SPECIFIC REMARKS

Title

Not clear. Homogeneous sporulation (what does that mean? Synchronic?). It is not the homogenous sporulation that is responsible for the spore modifications but the overexpression of kinA. Please change the title to make it more relevant to the results.

The main remarks concern the "Result part". I have therefore focused this review on the results, in relation to the methods used.

2/ Results (& corresponding sections in M&M)

Please, can you quickly remind the reader how the mutant was constructed and whether in M-, kinA is expressed as in wild strains or totally inhibited?

Section 2.1

Synchronization vs time sporulation

  1. In order to conclude on the synchronization of sporulation, additional data at other sporulation times would be required. For example, in the Figure 1A, there is no evidence that the sporulation of the wild strain is not synchronous but simply delayed as compared to the mutant.
  2. Actually, it might be interesting to clarify what the authors mean by synchronous: what is the expected sporulation rate, in how long (time between the appearance of the first and the last spore)? Indeed, in the text, it is unclear whether the authors are referring to the rapidity of spore production or to a truly synchronous effect of kinA.
  3. Not being an expert on the subject, I wonder if, with 70% of spores after 8 h, 80% after 24 h and maybe more after that, we can consider it to be synchronous sporulation. In other words, after one-day sporulation, 20% to 50% of the population is still in different physiological conditions (not in the form of spores), which can affect proteomic studies. What do other authors in the published literature consider a synchronous culture?

How do the authors explain the difference observed in the sporulation rate between M- and the other strains after 24 h?

Section 2.2

First of all, I have the same question as above. Is M- able to express KinA or not?

I think it would be useful to add some details here or in the material section to clarify this paragraph.

  • How do the authors measure the thickness of the different spore layers (highly variable, even at the level of a single spore): average values (or maximum values) from a given number of measurements on a single spore?
  • I think it is important to have images of the two strains with and without IPTG, et a lower magnification (to observe a few spores per view) to be able to clearly observe the possible variability among spores
  • Please indicate how many spore batches have been analyzed (here and in the rest of the "Result" section.
  • I was unable to find any information on the statistical analyses used in this study. Did the variance analysis take the spore batch into account? Or other parameters? If so, I would have liked the authors to also present the differences observed between spore batches. The number of spores tested varies between strains. Why? I would have preferred that the analyses be performed on the same number of spores so as not to affect the statistical analysis of the data.
  • Actually, I find the results of the statistical analyses very surprising in view of the results presented in Figure 2. Therefore, it might be more careful to describe the experimental results (e.g., most spores have a coat that measures between 60 and 100 nm) before to give the statistic result.

Section 2.3

As I understood it, the spores are harvested at 24 h. Are you sure that sporulation stopped after one day?

Elsewhere, it is now largely admitted that many spores could be only injured following a mild heat treatment (e.g. 85°C) and need some time or specific conditions to be able to form colonies. What I mean is that in the experimental conditions of the study, we only know that some spores became unable to form colonies in 2 days, but not that they survived or not the treatment. Which is already a result in itself.

Here, the more striking result is the difference observed between M+/- and WT+/- strains, yet the authors do not describe this result. What would be the reason for this increased resistance of M+/- strains? What are the differences potentially induced by the mutation, regardless of kinA expression?

Conversely the increased resistance of M+ was noticed by the authors. This being said, it was only shown after a 10-min treatment, not for longer treatments. how can the authors explain this phenomenon?

Line 35. What do you mean by end of germination?

Section 2.3

I think information on WT- are lacking, mainly to check why many proteins are differently expressed in M- as compared to M+ and WT+. Is it due to the addition of IPTG? Or the lack of kinA expression in M-?

Section 2.4

Concerning KinA, a huge amount of proteins are affected by its expression. Apart from listing the up- or down-regulated proteins, it is difficult to link their presence to any character of the spore or even the vegetative cell.

3/ Discussion

Most remarks have been already discussed in the "Result" sections.

The discussion focuses on the proteomic comparison of spores. In my opinion, it seems that the topic of the article has evolved between the introduction and the discussion. From now on, the synchronous character of sporulation is only a property like any other resulting from the overexpression of kinA.

As indicated in the "General remarks", perhaps the entire organization of this article should be reviewed to ensure that the objective is clearly defined and analyzed throughout the document.

Author Response

Concerns: rebuttal Induced homogeneous sporulation by kinA-overexpression leads to altered Bacillus subtilis spore proteomes and spore properties. IJMS-802450 (see also attachment).

 Dear editor,

I would hereby like to submit our rebuttal and amended manuscript. We would like to answer the points put forward by the reviewers as follows.

First, we agree with both reviewers that the title did not optimally match the content of the manuscript. Hence, we have revised the title to: Artificial sporulation induction (ASI) by KinA overexpression affects the proteomes and properties of Bacillus subtilis spores.

Below we answer the specific points put forward by reviewer 2, whom we thank for the constructive criticism.

Reviewer 2

The purpose of this study is not very clear. Is it to achieve synchronized sporulation or to analyze the consequences of over-expression or inhibition of kinA expression? Consequently, the entire organization of this article should be reviewed to ensure that the objective is clearly defined and analyzed throughout the document.

In addition, in my opinion, essential information is missing (see below various remarks in the "Result" section). As a result, the interpretation of some results can be controversial.

in conclusion, I consider that this article is not acceptable as it stands to be published in IJMS.

Our remark: While reviewer 1 was clearer on the intention that we had, we nonetheless recognize that both reviewers point out that the clarity of our objective was not appropriately reflected by the title of the study. Hence, we amended the title as outlined above and below.

SPECIFIC REMARKS

Title

Not clear. Homogeneous sporulation (what does that mean? Synchronic?). It is not the homogenous sporulation that is responsible for the spore modifications but the overexpression of kinA. Please change the title to make it more relevant to the results.

Our reply: In agreement with the reviewer’s comments the title has been modified to: “KinA overexpression affects the proteomes and properties of Bacillus subtilis spores prepared by artificial sporulation induction (ASI).”

The main remarks concern the "Result part". I have therefore focused this review on the results, in relation to the methods used.

2/ Results (& corresponding sections in M&M)

Please, can you quickly remind the reader how the mutant was constructed and whether in M-, kinA is expressed as in wild strains or totally inhibited?

Our reply: An explanation was added to the M&M section, line 294-296. The mutant strain was obtained from Prof. M. Fujita’s lab from University of Houston. Details about relevant studies doen with the mutant can be obtained from “Eswaramoorthy P, Duan D, Dinh J, Dravis A, Devi SN, Fujita M. The threshold level of the sensor histidine kinase KinA governs entry into sporulation in Bacillus subtilis. J Bacteriol. 2010;192(15):3870‐3882. doi:10.1128/JB.00466-10” and regarding its original construction from ”Fujita M. and Losick, R. Evidence that entry into sporulation in Bacillus subtilis is governed by a gradual increase in the level and activity of the master regulator Spo0A. Genes & Development 2005; 19: 2236-2244  doi/10.1101/gad.1335705”. The text is amended and reference added. When IPTG is absent, there is no kinA is expression in kinA-inducible mutant cells. Nevertheless, kinB and kinC rescue the sporulation process.

Section 2.1

Synchronization vs time sporulation

  1. In order to conclude on the synchronization of sporulation, additional data at other sporulation times would be required. For example, in the Figure 1A, there is no evidence that the sporulation of the wild strain is not synchronous but simply delayed as compared to the mutant.

Our reply: The existence of heterogeneity in sporulation is widespread and a well-established fact to the spore community, prominently involving studies of spore formation in Bacilli. Lines 49 – 62 give up-to-date information about this heterogeneity. What we achieve here is synchronized initiation of sporulation that leads to 70% spores in 8 hours, a time frame commonly seen as required for spores to form (see for instance “Veening, J.W., Smits, W.K., Hamoen, L.W. and Kuipers, O.P. Single cell analysis of gene expression patterns of competence development and initiation of sporulation in Bacillus subtilis grown on chemically defined media. J. Appl. Microbiol. 2006;101, 531-541. doi.org/10.1111/j.1365-2672.2006.02911.x”). Given the well-known heterogeneity in sporulation and our results, it is certain that only synchronized initiation can lead to such high numbers of spores in such a relatively short time (see also the remarks by reviewer 1). Hence, we do not think adding further time points to the manuscript will add new insights to the results. Still, we provide, for review purposes, the images of induced mutant cells at 3, 4, 5, 6 and 7 h post-induction. Please refer to the file ‘microscopic images’.

Actually, it might be interesting to clarify what the authors mean by synchronous: what is the expected sporulation rate, in how long (time between the appearance of the first and the last spore)? Indeed, in the text, it is unclear whether the authors are referring to the rapidity of spore production or to a truly synchronous effect of kinA.

Our reply: First, we would like to refer the reviewer to our rebuttal to his first point. Furthermore, in agreement with the reviewer, we would like to make it clear that we are trying to make the initiation of sporulation as much as possible synchronized at a certain timepoint and analyze its molecular physiological consequences for the resulting spores. Thus we aim at synchronous sporulation initiation which is now accordingly throughout the article.

  1. Not being an expert on the subject, I wonder if, with 70% of spores after 8 h, 80% after 24 h and maybe more after that, we can consider it to be synchronous sporulation. In other words, after one-day sporulation, 20% to 50% of the population is still in different physiological conditions (not in the form of spores), which can affect proteomic studies. What do other authors in the published literature consider a synchronous culture?

Our reply: In the published literature, a synchronous culture is defined as the one where all the cells in the culture are in the same physiological state at a given time. In our case, although there might still be 20%-50% of cells in the culture, they are removed using density gradient centrifugation to obtain >99% spores in the sample. (The free spores have a relative high density compared to vegetative cells and other sporulating cells.) This has been clarified M&M lines 313:316. One may operationally define the purified sample as homogeneous since there are only spores present in it. Still they may be spores of different age if the initiation of their formation was not synchronous, hence the efforts to come so such synchronous initiation to try to minimize every possible heterogeneity that could perturb proteomic analysis.   

How do the authors explain the difference observed in the sporulation rate between M- and the other strains after 24 h?

Our reply: Bacillus subtilis has five sensor kinases (KinA, KinB, KinC, KinD and KinE). KinA and KinB have experimental evidence to be involved in the initiation of sporulation, (line 54). When KinA is absent, KinB can rescue the sporulation process albeit likely with less efficiency. This has been clarified in the revised text (line 55 and lines 239-240).

Section 2.2

First of all, I have the same question as above. Is M- able to express KinA or not?

I think it would be useful to add some details here or in the material section to clarify this paragraph.

Our reply: As explained in M&M lines 297-299 no KinA is expressed when there is no IPTG provided to the mutant cells.

How do the authors measure the thickness of the different spore layers (highly variable, even at the level of a single spore): average values (or maximum values) from a given number of measurements on a single spore?

Our reply: The text in M&M lines 333-335 is amended. The thickness of different spore layers is an average of four measurements at different positions on the rectangular cross- sections for each spore and for each layer. Diameters of spores and spore cores are the averages of two measurements across horizontal and vertical axes.

I think it is important to have images of the two strains with and without IPTG, at a lower magnification (to observe a few spores per view) to be able to clearly observe the possible variability among spores.

Our reply: Upon the reviewer’s request, we have added EM images form M+/- and WT+/- spore in the supplemental material. Please refer to the file ‘supplementary file 1’.

Please indicate how many spore batches have been analyzed (here and in the rest of the "Result" section.

Our reply: We used one batch of spores for EM analysis. Three biological replicates were performed to check the sporulation efficiency, for live imaging analysis, for heat resistance tests and for proteomics analysis. This is clarified in the revised text.

I was unable to find any information on the statistical analyses used in this study. Did the variance analysis take the spore batch into account? Or other parameters? If so, I would have liked the authors to also present the differences observed between spore batches. The number of spores tested varies between strains. Why? I would have preferred that the analyses be performed on the same number of spores so as not to affect the statistical analysis of the data.

Our reply: We intended to have a similar number of spores in all analyses. However, not all spores could be imaged properly or measured due to technical limitations. Therefore, we have aimed to include at least 50 spores, per condition, for our analyses.

Actually, I find the results of the statistical analyses very surprising in view of the results presented in Figure 2. Therefore, it might be more careful to describe the experimental results (e.g., most spores have a coat that measures between 60 and 100 nm) before to give the statistic result.

Our reply: In agreement with the reviewer the text was amended along these lines in describing the experimental results.

Section 2.3

As I understood it, the spores are harvested at 24 h. Are you sure that sporulation stopped after one day?

Our reply: Owing to the inherent heterogeneity some non-sporulated cells are expected to be present in the culture after 24 hours but those are separated from the spores during gradient centrifugation. Furthermore, post-harvest the samples are stored at -80 ℃, so no further sporulation may occur in our harvested samples.

Elsewhere, it is now largely admitted that many spores could be only injured following a mild heat treatment (e.g. 85°C) and need some time or specific conditions to be able to form colonies. What I mean is that in the experimental conditions of the study, we only know that some spores became unable to form colonies in 2 days, but not that they survived or not the treatment. Which is already a result in itself.

Our reply: In agreement with the reviewer’s point about the heat resistance tests, we cannot be certain if the spores that do not form the colonies are dead or only partially damaged. However, the colonies are counted 48 hours after plating on a rich medium, until when the damaged spores might recover to form colonies. Since this is a routine test in the spore field, we are confident about our results regarding spore thermal treatment survival.

Here, the more striking result is the difference observed between M+/- and WT+/- strains, yet the authors do not describe this result. What would be the reason for this increased resistance of M+/- strains? What are the differences potentially induced by the mutation, regardless of kinA expression?

Our reply: The M+ spores have some upregulated coat proteins and thus thicker coat and cortex layers compared with M- and WT+ spores. We think that the role played by the spore cortex layer and the spore coat might be important in this case. A reason for such changes might be that the promoter used is slightly leaky, thus allowing a minimal basal level of expression. Still, see also our reply to reviewer 1, such leakiness was never reported before. Hence more research is needed to unravel the mechanisms behind the observed molecular changes.

Conversely the increased resistance of M+ was noticed by the authors. This being said, it was only shown after a 10-min treatment, not for longer treatments. how can the authors explain this phenomenon?

Our reply: Thermal treatments for varying times have cumulative effects on living systems. Hence, one possibility is that a longer heat treatment causes more severe damage to the spores making them unable to repair the damage and form colonies.

Line 35. What do you mean by end of germination?

Our reply: We think the reviewer refers to line 135. End of germination is the completion of phase-darkening.

Section 2.3

I think information on WT- are lacking, mainly to check why many proteins are differently expressed in M- as compared to M+ and WT+. Is it due to the addition of IPTG? Or the lack of kinA expression in M-?

Our reply: In result section 2.3 we do have results of WT- spores.

We have compared proteomes of WT+ and WT- spores (lines 197-200). Only two proteins are differently expressed, and they are not related to sporulation. Since WT+ and WT- have a difference of only two proteins, we think there is no necessity to compare M+ and WT-.

Regarding the differences in M-, we cannot rule out the polar effects of the mutation. Based on the article, ‘Eswaramoorthy P, Duan D, Dinh J, Dravis A, Devi SN, Fujita M. The threshold level of the sensor histidine kinase KinA governs entry into sporulation in Bacillus subtilis. J Bacteriol. 2010;192(15):3870‐3882. doi:10.1128/JB.00466-10’, where the mutant was extensively used and ‘Fujita M. and Losick, R. Evidence that entry into sporulation in Bacillus subtilis is governed by a gradual increase in the level and activity of the master regulator Spo0A. Genes & Development 2005; 19: 2236-2244  doi/10.1101/gad.1335705’, where the mutant was originally described, said mutation is leak proof. Therefore, we also think the same. But KinA overexpression affects many regulons as shown in the results. Therefore, there may be some regulatory link to this effect. Future more detailed studies will be needed to shed more light on this observation.

Section 2.4

Concerning KinA, a huge amount of proteins are affected by its expression. Apart from listing the up- or down-regulated proteins, it is difficult to link their presence to any character of the spore or even the vegetative cell.

Our reply: In discussion lines 241-266, we discuss the relationship between protein levels and the high thermal resistance, delayed germination as well as larger spore size. The relationship can be concluded as overexpression of KinA resulting in some coat proteins being upregulated, further increasing the thickness of coat and cortex layers, and ultimately raising their resistance to wet heat. Overexpression of KinA also causes downregulation of GerBC, one of the germinant receptor proteins. The protein amount of germinant receptors has a direct impact on the ability of spores to revive to vegetative cells. Moreover, the downregulation of GerBC could affect the predicted interaction with GerA receptor (https://www.ncbi.nlm.nih.gov/pmc/articles/PMC3147670/) thereby affecting germination. This could partially explain why KinA induced spores are delayed in germination. 

3/ Discussion

Most remarks have been already discussed in the "Result" sections.

The discussion focuses on the proteomic comparison of spores. In my opinion, it seems that the topic of the article has evolved between the introduction and the discussion. From now on, the synchronous character of sporulation is only a property like any other resulting from the overexpression of kinA.

As indicated in the "General remarks", perhaps the entire organization of this article should be reviewed to ensure that the objective is clearly defined and analyzed throughout the document.

Our reply: Upon the reviewer’s suggestion the text in the article is amended for clarification.